# LABEL-EFFICIENT SEMANTIC SEGMENTATION WITH DIFFUSION MODELS

**Dmitry Baranchuk, Ivan Rubachev, Andrey Voynov, Valentin Khrulkov, Artem Babenko**

Yandex Research

## ABSTRACT

Denoising diffusion probabilistic models have recently received much research attention since they outperform alternative approaches, such as GANs, and currently provide state-of-the-art generative performance. The superior performance of diffusion models has made them an appealing tool in several applications, including inpainting, super-resolution, and semantic editing. In this paper, we demonstrate that diffusion models can also serve as an instrument for semantic segmentation, especially in the setup when labeled data is scarce. In particular, for several pretrained diffusion models, we investigate the intermediate activations from the networks that perform the Markov step of the reverse diffusion process. We show that these activations effectively capture the semantic information from an input image and appear to be excellent pixel-level representations for the segmentation problem. Based on these observations, we describe a simple segmentation method, which can work even if only a few training images are provided. Our approach significantly outperforms the existing alternatives on several datasets for the same amount of human supervision. The source code of the project is publicly available.

## 1 INTRODUCTION

Denoising diffusion probabilistic models (DDPM) (Sohl-Dickstein et al., 2015; Ho et al., 2020) have recently outperformed alternative approaches to model the distribution of natural images both in the realism of individual samples and their diversity (Dhariwal & Nichol, 2021). These advantages of DDPM are successfully exploited in applications, such as colorization (Song et al., 2021), inpainting (Song et al., 2021), super-resolution (Saharia et al., 2021; Li et al., 2021b), and semantic editing (Meng et al., 2021), where DDPM often achieve more impressive results compared to GANs.

So far, however, DDPM were not exploited as a source of effective image representations for discriminative computer vision problems. While the prior literature has demonstrated that various generative paradigms, such as GANs (Donahue & Simonyan, 2019) or autoregressive models (Chen et al., 2020a), can be used to extract the representations for common vision tasks, it is not clear if DDPM can also serve as representation learners. In this paper, we provide an affirmative answer to this question in the context of semantic segmentation.

In particular, we investigate the intermediate activations from the U-Net network that approximates the Markov step of the reverse diffusion process in DDPM. Intuitively, this network learns to denoise its input, and it is not clear why the intermediate activations should capture semantic information needed for high-level vision problems. Nevertheless, we show that on certain diffusion steps, these activations do capture such information, and therefore, can potentially be used as image representations for downstream tasks. Given these observations, we propose a simple semantic segmentation method, which exploits these representations and works successfully even if only a few labeled images are provided. On several datasets, we show that our DDPM-based segmentation method outperforms the existing baselines for the same amount of supervision.

To sum up, the contributions of our paper are:

1. We investigate the representations learned by the state-of-the-art DDPM and show that they capture high-level semantic information valuable for downstream vision tasks.

2. We design a simple semantic segmentation approach that exploits these representations and outperforms the alternatives in the few-shot operating point.

3. We compare the DDPM-based representations with their GAN-based counterparts on the same datasets and demonstrate the advantages of the former in the context of semantic segmentation.

## 2 RELATED WORK

In this section, we briefly describe the existing lines of research relevant to our work.

**Diffusion models** (Sohl-Dickstein et al., 2015; Ho et al., 2020) are a class of generative models that approximate the distribution of real images by the endpoint of the Markov chain which originates from a simple parametric distribution, typically a standard Gaussian. Each Markov step is modeled by a deep neural network that effectively learns to invert the diffusion process with a known Gaussian kernel. Ho et al. highlighted the equivalence of diffusion models and score matching (Song & Ermon, 2019; 2020), showing them to be two different perspectives on the gradual conversion of a simple known distribution into a target distribution via the iterative denoising process. Very recent works (Nichol, 2021; Dhariwal & Nichol, 2021) have developed more powerful model architectures as well as different advanced objectives, which led to the "victory" of DDPM over GANs in terms of generative quality and diversity. DDPM have been widely used in several applications, including image colorization (Song et al., 2021), super-resolution (Saharia et al., 2021; Li et al., 2021b), in-painting (Song et al., 2021), and semantic editing (Meng et al., 2021). In our work, we demonstrate that one can also successfully use them for semantic segmentation.

**Image segmentation with generative models** is an active research direction at the moment, however, existing methods are primarily based on GANs. The first line of works (Voynov & Babenko, 2020; Voynov et al., 2021; Melas-Kyriazi et al., 2021) is based on the evidence that the latent spaces of the state-of-the-art GANs have directions corresponding to effects that influence the foreground/background pixels differently, which allows producing synthetic data to train segmentation models. However, these approaches are currently able to perform binary segmentation only, and it is not clear if they can be used in the general setup of semantic segmentation. The second line of works (Zhang et al., 2021; Tritrong et al., 2021; Xu, 2021; Galeev et al., 2020) is more relevant to our study since they are based on the intermediate representations obtained in GANs. In particular, the method proposed in (Zhang et al., 2021) trains a pixel class prediction model on these representations and confirms their label efficiency. In the experimental section, we compare the method from (Zhang et al., 2021) to our DDPM-based one and demonstrate several distinctive advantages of our solution.

**Representations from generative models for discriminative tasks.** The usage of generative models, as representation learners, has been widely investigated for global prediction (Donahue & Simonyan, 2019; Chen et al., 2020a), and dense prediction problems (Zhang et al., 2021; Tritrong et al., 2021; Xu, 2021; Xu et al., 2021). While previous works highlighted the practical advantages of these representations, such as out-of-distribution robustness (Li et al., 2021a), generative models as representation learners receive less attention compared to alternative unsupervised methods, e.g., based on contrastive learning (Chen et al., 2020b). The main reason is probably the difficulty of training a high-quality generative model on a complex, diverse dataset. However, given the recent success of DDPM on Imagenet (Deng et al., 2009), one can expect that this direction will attract more attention in the future.

## 3 REPRESENTATIONS FROM DIFFUSION MODELS

In the following section, we investigate the image representations learned by diffusion models. First, we provide a brief overview of the DDPM framework. Then, we describe how to extract features with DDPM and investigate what kind of semantic information these features might capture.

**Background.** Diffusion models transform noise $x_T \sim N(0, I)$ to the sample $x_0$ by gradually denoising $x_T$ to less noisy samples $x_t$. Formally, we are given a forward diffusion process:

$$q(x_t|x_{t-1}) := \mathcal{N}(x_t; \sqrt{1 - \beta_t}x_{t-1}, \beta_t I), \tag{1}$$

for some fixed variance schedule $\beta_1, \ldots, \beta_t$.

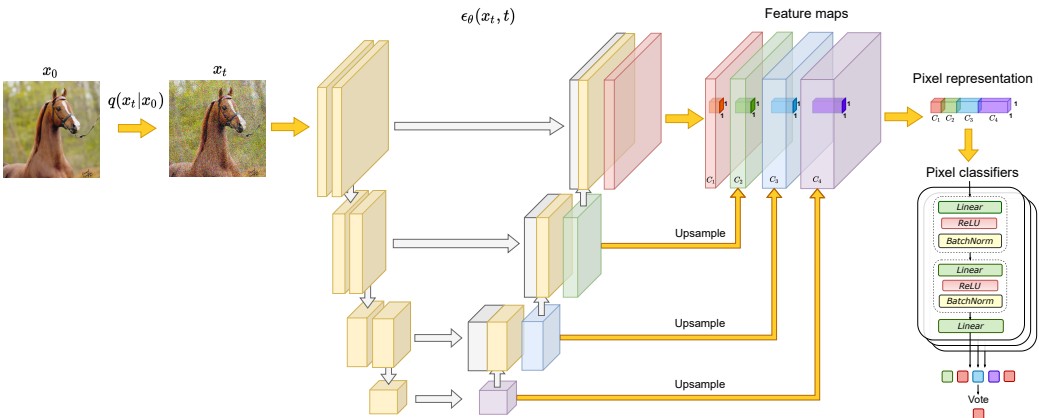

Figure 1: **Overview of the proposed method.** (1) $x_0 \rightarrow x_t$ by adding noise according to $q(x_t|x_0)$. (2) Extracting feature maps from a noise predictor $\epsilon_\theta(x_t, t)$. (3) Collecting pixel-level representations by upsampling the feature maps to the image resolution and concatenating them. (4) Using the pixel-wise feature vectors to train an ensemble of MLPs to predict a class label for each pixel.

Importantly, a noisy sample $x_t$ can be obtained directly from the data $x_0$:

$$q(x_t|x_0) := \mathcal{N}(x_t; \sqrt{\bar{\alpha}_t}x_0, (1 - \bar{\alpha}_t)I),$$
$$x_t = \sqrt{\bar{\alpha}_t}x_0 + \sqrt{1 - \bar{\alpha}_t}\epsilon, \quad \epsilon \sim \mathcal{N}(0, 1), \tag{2}$$

where $\alpha_t := 1 - \beta_t$, $\bar{\alpha}_t := \prod_{s=1}^{t} \alpha_s$.

Pretrained DDPM approximates a reverse process:

$$p_\theta(x_{t-1}|x_t) := \mathcal{N}(x_{t-1}; \mu_\theta(x_t, t), \Sigma_\theta(x_t, t)). \tag{3}$$

In practice, rather than predicting the mean of the distribution in Equation (3), the noise predictor network $\epsilon_\theta(x_t, t)$ predicts the noise component at the step $t$; the mean is then a linear combination of this noise component and $x_t$. The covariance predictor $\Sigma_\theta(x_t, t)$ can be either a fixed set of scalar covariances or learned as well (the latter was shown to improve the model quality (Nichol, 2021)).

The denoising model $\epsilon_\theta(x_t, t)$ is typically parameterized by different variants of the UNet architecture (Ronneberger et al., 2015), and in our experiments we investigate the state-of-the-art one proposed in (Dhariwal & Nichol, 2021).

**Extracting representations.** For a given real image $x_0 \in \mathbb{R}^{H \times W \times 3}$, one can compute $T$ sets of activation tensors from the noise predictor network $\epsilon_\theta(x_t, t)$. The overall scheme for a timestep $t$ is presented in Figure 1. First, we corrupt $x_0$ by adding Gaussian noise according to Equation (2). The noisy $x_t$ is used as an input of $\epsilon_\theta(x_t, t)$ parameterized by the UNet model. The UNet's intermediate activations are then upsampled to $H \times W$ with bilinear interpolation. This allows treating them as pixel-level representations of $x_0$.

### 3.1 REPRESENTATION ANALYSIS

We analyze the representations produced by the noise predictor $\epsilon_\theta(x_t, t)$ for different $t$. We consider the state-of-the-art DDPM checkpoints trained on the LSUN-Horse and FFHQ-256 datasets[1].

**The intermediate activations from the noise predictor capture semantic information.** For this experiment, we take a few images from the LSUN-Horse and FFHQ datasets and manually assign each pixel to one of the 21 and 34 semantic classes, respectively. Our goal is to understand whether the pixel-level representations produced by DDPM effectively capture the information about semantics. To this end, we train a multi-layer perceptron (MLP) to predict the pixel semantic label from its features produced by one of the 18 UNet decoder blocks on a specific diffusion step $t$. Note that we consider only the decoder activations because they also aggregate the encoder activations through the skip connections. MLPs are trained on 20 images and evaluated on 20 hold-out ones. The predictive performance is measured in terms of mean IoU.

[1] https://github.com/openai/guided-diffusion

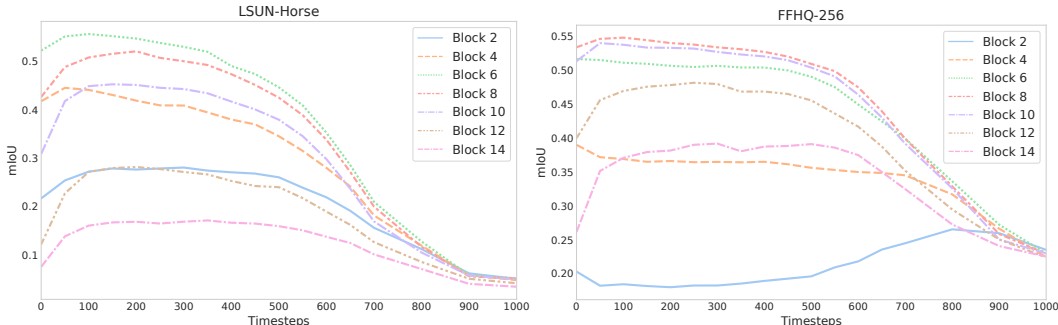

Figure 2: The evolution of predictive performance of DDPM-based pixel-wise representations for different UNet decoder blocks and diffusion steps. The blocks are numbered from the deep to shallow ones. The most informative features typically correspond to the later steps of the reverse diffusion process and middle layers of the UNet decoder. The earlier steps correspond to uninformative representations. The plots for other datasets are provided in Appendix A

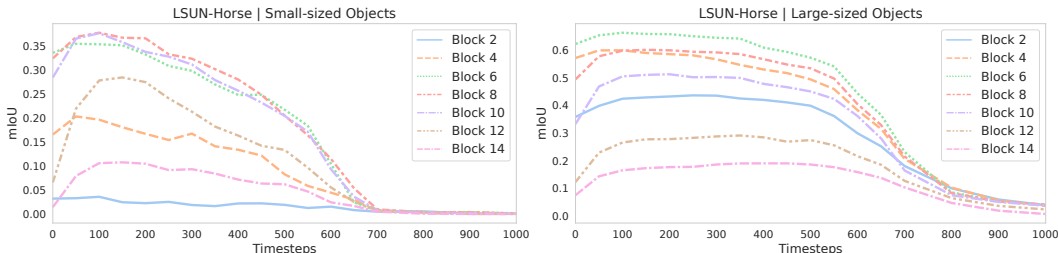

Figure 3: The evolution of predictive performance of DDPM-based pixel-wise representations on the LSUN-Horse dataset for classes with the smallest (**Left**) and largest (**Right**) average areas. The predictive performance for small-sized objects starts growing later in the reverse process. The deeper blocks are more informative for larger objects and the shallower blocks are more informative for smaller objects. A similar evaluation for other datasets is provided in Appendix A.

The evolution of predictive performance across the different blocks and diffusion steps $t$ is presented in Figure 2. The blocks are numbered from the deep to shallow ones. Figure 2 shows that the discriminability of the features produced by the noise predictor $\epsilon_\theta(x_t, t)$ varies for different blocks and diffusion steps. In particular, the features corresponding to the later steps of the reverse diffusion process typically capture semantic information more effectively. In contrast, the ones corresponding to the early steps are generally uninformative. Across different blocks, the features produced by the layers in the middle of the UNet decoder appear to be the most informative on all diffusion steps.

Also, we separately consider small-sized and large-sized semantic classes based on the average area in the annotated dataset. Then, we evaluate mean IoU for these classes independently across the different UNet blocks and diffusion steps. The results on LSUN-Horse are in Figure 3. As expected, the predictive performance for large-sized objects starts growing earlier in the reverse process. The shallower blocks are more informative for smaller objects, while the deeper blocks are more so for the larger ones. In both cases, the most discriminative features still correspond to the middle blocks.

Figure 2 implies that for certain UNet blocks and diffusion steps, similar DDPM-based representations correspond to the pixels of the same semantics. Figure 4 shows the k-means clusters ($k$=5) formed by the features extracted by the FFHQ checkpoint from the blocks $\{6, 8, 10, 12\}$ on the diffusion steps $\{50, 200, 400, 600, 800\}$, and confirms that clusters can span coherent semantic objects and object-parts. In the block $B$=6, the features correspond to coarse semantic masks. At the other extreme, the features from $B$=12 can discriminate between fine-grained face parts but exhibit less semantic meaningness for coarse fragmentation. Across different diffusion steps, the most meaningful features correspond to the later ones. We attribute this behavior to the fact that on the earlier steps of the reverse process, the global structure of a DDPM sample has not yet emerged, therefore, it is hardly possible to predict segmentation masks at this stage. This intuition is qualitatively confirmed by the masks in Figure 4. For $t$=800, the masks poorly reflect the content of actual images, while for smaller values of $t$, the masks and images are semantically coherent.

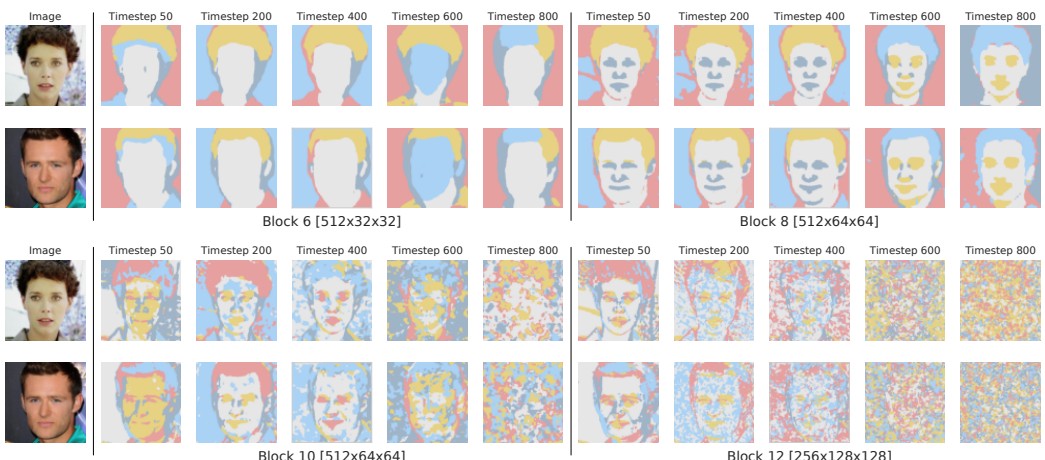

Figure 4: Examples of k-means clusters ($k$=5) formed by the features extracted from the UNet decoder blocks $\{6, 8, 10, 12\}$ on the diffusion steps $\{50, 200, 400, 600, 800\}$. The clusters from the middle blocks spatially span coherent semantic objects and parts.

## 3.2 DDPM-BASED REPRESENTATIONS FOR FEW-SHOT SEMANTIC SEGMENTATION

The potential effectiveness of the intermediate DDPM activations observed above implies their usage as image representations for dense prediction tasks. Figure 1 schematically presents our overall approach for image segmentation, which exploits the discriminability of these representations. In more detail, we consider a few-shot semi-supervised setup, when a large number of unlabeled images $\{X_1, \ldots, X_N\} \subset \mathbb{R}^{H \times W \times 3}$ from the particular domain are available, and only for $n$ training images $\{X_1, \ldots, X_n\} \subset \mathbb{R}^{H \times W \times 3}$ the groundtruth $K$-class semantic masks $\{Y_1, \ldots, Y_n\} \subset \mathbb{R}^{H \times W \times \{1, \ldots, K\}}$ are provided.

As a first step, we train a diffusion model on the whole $\{X_1, \ldots, X_N\}$ in an unsupervised manner. Then, this diffusion model is used to extract the pixel-level representations of the labeled images using the subset of the UNet blocks and diffusion steps $t$. In this work, we use the representations from the middle blocks $B=\{5, 6, 7, 8, 12\}$ of the UNet decoder and later steps $t=\{50, 150, 250\}$ of the reverse diffusion process. These blocks and time steps are motivated by the insights from Section 3.1 but intentionally not tuned for each dataset.

While the feature extraction at the particular time step is stochastic, we fix the noise for all timesteps $t$ and ablate this in Section 4.1. The extracted representations from all blocks $B$ and steps $t$ are upsampled to the image size and concatenated, forming the feature vectors for all pixels of the training images. The overall dimension of the pixel-level representations is $8448$.

Then, following (Zhang et al., 2021), we train an ensemble of independent multi-layer perceptrons (MLPs) on these feature vectors, which aim to predict a semantic label of each pixel available for training images. We adopt the ensemble configuration and training settings from (Zhang et al., 2021) and exploit them across all other methods in our experiments, see Appendix C for details.

To segment a test image, we extract its DDPM-based pixel-wise representations and use them to predict the pixel labels by the ensemble. The final prediction is obtained by majority voting.

## 4 EXPERIMENTS

This section experimentally confirms the advantage of the DDPM-based representations for the semantic segmentation problem. We start from a thorough comparison to the existing alternatives and then dissect the reasons for the DDPM success by additional analysis.

**Datasets.** In our evaluation, we mainly work with the "bedroom", "cat" and "horse" categories from LSUN (Yu et al., 2015) and FFHQ-256 (Karras et al., 2019). As a training set for each dataset, we consider several images for which the fine-grained semantic masks are collected following the protocol from (Zhang et al., 2021). For each dataset, a professional assessor was hired to annotate train and test samples. We denote the collected datasets as **Bedroom-28**, **FFHQ-34**, **Cat-15**, **Horse-21**, where the number corresponds to the number of semantic classes.

| Dataset | Real$_{Train}$ | Real$_{Test}$ | GAN | DDPM | Total |
|---|---|---|---|---|---|
| Bedroom-28 | 40 | 20 | 40 | 40 | 140 |
| FFHQ-34 | 20 | 20 | 20 | 20 | 80 |
| Cat-15 | 30 | 20 | 30 | 30 | 110 |
| Horse-21 | 30 | 30 | 30 | 30 | 120 |
| CelebA-19 | 20 | 500 | — | — | 520 |
| ADE-Bedroom-30 | 50 | 650 | — | — | 700 |

Table 1: Number of annotated images for each dataset used in our evaluation.

Additionally, we consider two datasets, which, in contrast to others, have publicly available annotations and sizable evaluation sets:

- **ADE-Bedroom-30** is a subset of the ADE20K dataset (Zhou et al., 2018), where we extract only images of bedroom scenes with 30 most frequent classes. We resize each image to 256 for the smaller side and then crop them to obtain the $256 \times 256$ samples.

- **CelebA-19** is a subset of the CelebAMask-HQ dataset (Lee et al., 2020), which provides the annotation for 19 facial attributes. All images are resized to 256 resolution.

The number of annotated images for each dataset are in Table 1. Other details are in Appendix E.

**Methods.** In the evaluation, we compare our method (denoted as **DDPM**) to several prior approaches which tackle the few-shot semantic segmentation setup. First, we describe the baselines that produce a large set of annotated synthetic images to train a segmentation model:

- **DatasetGAN** (Zhang et al., 2021) — this method exploits the discriminability of pixel-level features produced by GANs. In more detail, assessors annotate a few GAN-produced images. Then, the latent codes of these images are used to obtain the intermediate generator activations, which are considered as pixel-level representations. Given these representations, a classifier is trained to predict a semantic label for each pixel. This classifier is then used to label new synthetic GAN images, which, for their part, serve as a training set for the DeepLabV3 segmentation model (Chen et al., 2017). For each dataset, we increase the number of synthetic images until the performance on the validation set is not saturated. According to (Zhang et al., 2021), we also remove 10% of synthetic samples with the most uncertain predictions.

- **DatasetDDPM** mirrors the **DatasetGAN** baseline with the only difference being that GANs are replaced with DDPMs. We include this baseline to compare the GAN-based and DDPM-based representations in the same scenario.

Note that our segmentation method described in Section 3.2 is more straightforward compared to **DatasetGAN** and **DatasetDDPM** since it does not require auxiliary steps of the synthetic dataset generation and training the segmentation model on it.

Then, we consider a set of baselines that allow extracting intermediate activations from the real images directly and use them as pixel-level representations similarly to our method. In contrast to DatasetGAN and DatasetDDPM, these methods can potentially be beneficial due to the absence of the domain gap between real and synthetic images.

- **MAE** (He et al., 2021) — one of the state-of-the-art self-supervised methods, which learns a denoising autoencoder to reconstruct missing patches. We use ViT-Large (Dosovitskiy et al., 2021) as a backbone model and reduce the patch size to $8 \times 8$ to increase the spatial dimensions of the feature maps. We pretrain all models on the same datasets as DDPM using the official code[2]. The feature extraction for this method is described in Appendix F.

- **SwAV** (Caron et al., 2020) — one more recent self-supervised approach. We consider a twice wider ResNet-50 model for evaluation. All models are pretrained on the same datasets as DDPM also using the official source code[3]. The input image resolution is 256.

- **GAN Inversion** employs the state-of-the-art method (Tov et al., 2021) to obtain the latent codes for real images. We map the annotated real images to the GAN latent space, which allows computing the intermediate generator activations and using them as pixel-level representations.

---

[2]https://github.com/facebookresearch/mae
[3]https://github.com/facebookresearch/swav

| Method | Bedroom-28 | FFHQ-34 | Cat-15 | Horse-21 | CelebA-19* | ADE Bedroom-30* |
|---|---|---|---|---|---|---|
| ALAE | $20.0 \pm 1.0$ | $48.1 \pm 1.3$ | — | — | $49.7 \pm 0.7$ | $15.0 \pm 0.5$ |
| VDVAE | — | $57.3 \pm 1.1$ | — | — | $54.1 \pm 1.0$ | — |
| GAN Inversion | $13.9 \pm 0.6$ | $51.7 \pm 0.8$ | $21.4 \pm 1.7$ | $17.7 \pm 0.4$ | $51.5 \pm 2.3$ | $11.1 \pm 0.2$ |
| GAN Encoder | $22.4 \pm 1.6$ | $53.9 \pm 1.3$ | $32.0 \pm 1.8$ | $26.7 \pm 0.7$ | $53.9 \pm 0.8$ | $15.7 \pm 0.3$ |
| SwAV | $42.4 \pm 1.7$ | $56.9 \pm 1.3$ | $45.1 \pm 2.1$ | $54.0 \pm 0.9$ | $52.4 \pm 1.3$ | $30.6 \pm 1.6$ |
| MAE | $45.0 \pm 2.0$ | $\mathbf{58.8 \pm 1.1}$ | $\mathbf{52.4 \pm 2.3}$ | $63.4 \pm 1.4$ | $57.8 \pm 0.4$ | $31.7 \pm 1.8$ |
| DatasetGAN | $31.3 \pm 2.3$ | $57.0 \pm 1.1$ | $36.5 \pm 2.3$ | $45.4 \pm 1.4$ | — | — |
| DatasetDDPM (Ours) | $47.9 \pm 2.9$ | $56.0 \pm 0.9$ | $47.6 \pm 1.5$ | $60.8 \pm 1.0$ | — | — |
| **DDPM (Ours)** | $\mathbf{49.4 \pm 1.9}$ | $\mathbf{59.1 \pm 1.4}$ | $\mathbf{53.7 \pm 3.3}$ | $\mathbf{65.0 \pm 0.8}$ | $\mathbf{59.9 \pm 1.0}$ | $\mathbf{34.6 \pm 1.7}$ |

Table 2: The comparison of the segmentation methods in terms of mean IoU. (*) On CelebA-19 and ADE Bedroom-30, we evaluate models trained on FFHQ-256 and LSUN Bedroom, respectively.

- **GAN Encoder** — while GAN Inversion struggles to reconstruct images from LSUN domains, we also consider the activations of the pretrained GAN encoder used for GAN Inversion.

- **VDVAE** (Child, 2021) — state-of-the-art autoencoder model. The intermediate activations are extracted from both encoder and decoder and concatenated. While there are no pretrained models on the LSUN datasets, we evaluate this model only on the publicly available checkpoint[4] on FFHQ-256. Note that VAEs are still significantly inferior to GANs and DDPMs on LSUN.

- **ALAE** (Pidhorskyi et al., 2020) adopts StyleGANv1 generator and adds an encoder network to the adversarial training. We extract features from the encoder model. In our evaluation, we use publicly available models on LSUN-Bedroom and FFHQ-1024[5].

**Generative pretrained models.** In our experiments, we use the state-of-the-art StyleGAN2 (Karras et al., 2020) models for the GAN-based baselines and the state-of-the-art pretrained ADMs (Dhariwal & Nichol, 2021) for our DDPM-based method. Since there is not a pretrained model for FFHQ-256, we train it ourselves using the official implementation[6]. For evaluation on the ADE-Bedroom-30 dataset, we use the models (including the baselines) pretrained on LSUN-Bedroom. For Celeba-19, we evaluate the models trained on FFHQ-256.

**Main results.** The comparison of the methods in terms of the mean IoU measure is presented in Table 2. The results are averaged over 5 independent runs for different data splits. We also report per class IoUs in Appendix D. Additionally, we provide several qualitative examples of segmentation with our method in Figure 5. Below we highlight several key observations:

- The proposed method based on the DDPM representations significantly outperforms the alternatives on most datasets.

- The **MAE** baseline is the strongest competitor to the DDPM-based segmentation and demonstrates comparable results on the FFHQ-34 and Cat-15 datasets.

- The **SwAV** baseline underperforms compared to the DDPM-based segmentation. We attribute this behavior to the fact that this baseline is trained in the discriminative fashion and can suppress the details, which are needed for fine-grained semantic segmentation. This result is consistent with the recent findings in (Cole et al., 2021), which shows that the state-of-the-art contrastive methods produce representations, which are suboptimal for fine-grained problems.

- **DatasetDDPM** outperforms its counterpart **DatasetGAN** against most benchmarks. Note that both these methods use the DeepLabV3 network. We attribute this superiority to the higher quality of DDPM synthetics, therefore, a smaller domain gap between synthetic and real data.

- On most datasets, **DDPM** outperforms the **DatasetDDPM** competitor. We provide an additional experiment to investigate this in the discussion section below.

Overall, the proposed DDPM-based segmentation outperforms the baselines that exploit alternative generative models and also the baselines trained in the self-supervised fashion. This result highlights the potential of using the state-of-the-art DDPMs as strong unsupervised representation learners.

---

[4] https://github.com/openai/vdvae
[5] https://github.com/podgorskiy/ALAE
[6] https://github.com/openai/guided-diffusion

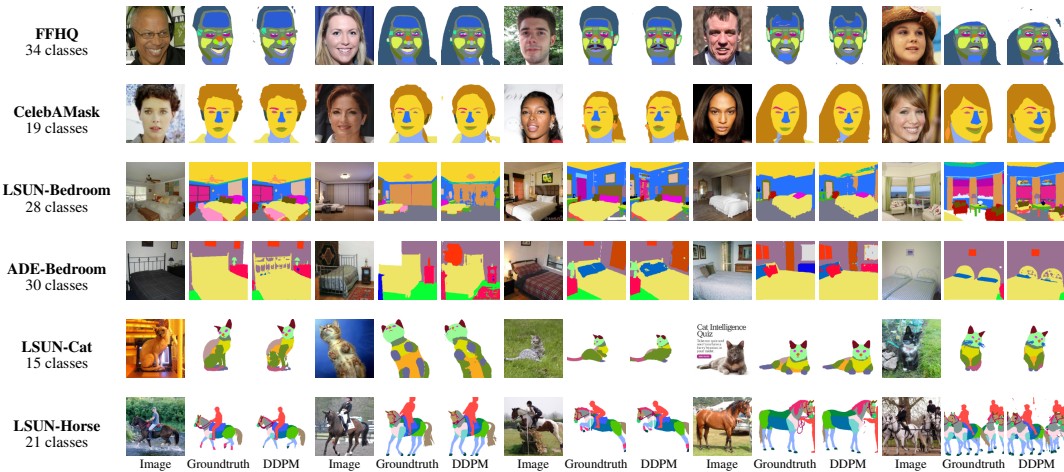

Figure 5: The examples of segmentation masks predicted by our method on the test images along with the groundtruth annotated masks.

| | Bedroom-28 | | | Cat-15 | | | Horse-21 | | |
|---|---|---|---|---|---|---|---|---|---|
| Train data | Real | DDPM | GAN | Real | DDPM | GAN | Real | DDPM | GAN |
| DatasetGAN | — | — | $31.3 \pm 2.3$ | — | — | $36.5 \pm 2.3$ | — | — | $45.4 \pm 1.4$ |
| DatasetDDPM | — | $47.9 \pm 2.9$ | — | — | $47.6 \pm 1.5$ | — | — | $60.8 \pm 1.0$ | — |
| DDPM | $49.4 \pm 1.9$ | $48.7 \pm 2.6$ | $43.3 \pm 2.9$ | $53.7 \pm 3.3$ | $47.9 \pm 2.7$ | $41.1 \pm 2.2$ | $65.0 \pm 0.8$ | $62.4 \pm 1.0$ | $60.0 \pm 1.0$ |

Table 3: Performance of DDPM-based segmentation when trained on real and synthetic images. When trained on DDPM-produced data, DDPM demonstrates comparable performance to Dataset-DDPM. When trained on GAN-produced data, DDPM still significantly outperforms DatasetGAN, but the gap between them reduces.

## 4.1 DISCUSSION

**The effect of training on real data**. The proposed DDPM method is trained on annotated real images, while DatasetDDPM and DatasetGAN are trained on synthetic ones, which are typically less natural, diverse, and can lack objects of particular classes. Moreover, synthetic images are harder for human annotation since they might have some distorted objects that are difficult to assign to a particular class. In the following experiment, we quantify the performance drop caused by training on real or synthetic data. Specifically, Table 3 reports the performance of the DDPM approach trained on real, DDPM-produced and GAN-produced annotated images. As can be seen, training on real images is very beneficial on the domains where the fidelity of generative models is still relatively low, e.g., LSUN-Cat, which indicates that annotated real images are a more reliable source of supervision. Moreover, if the DDPM method is trained on synthetic images, its performance becomes on par with DatasetDDPM. On the other hand, when trained on GAN-produced samples, DDPM significantly outperforms DatasetGAN. We attribute this to the fact that DDPMs provide more semantically-valuable pixel-wise representations compared to GANs.

**Sample-efficiency.** In this experiment, we evaluate the performance of our method when it utilizes less annotated data. We provide mIoU for four datasets in Table 4. Importantly, DDPM is still able to outperform most baselines in Table 2, using significantly less supervision.

**The effect of stochastic feature extraction.** Here, we investigate whether our method can benefit from the stochastic feature extraction described in Section 3.2. We consider the deterministic case, when the noise $\epsilon \sim N(0, I)$ is sampled once and used in (2) to obtain $x_t$ for all timesteps $t$ during both training and evaluation. Then, we compare it to the following stochastic options:

First, different $\epsilon_t$ are sampled for different timesteps $t$ and shared during the training and evaluation. Second, one samples different noise for all timesteps at each training iteration; during the evaluation the method also uses unseen noise samples.

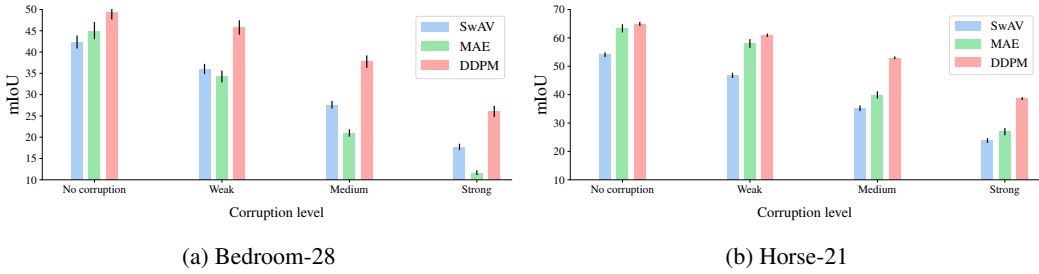

|  | (a) Bedroom-28 | (b) Horse-21 |
|---|---|---|

Figure 6: mIoU degradation for different image corruption levels on the Bedroom-28 and Horse-21 datasets. DDPM demonstrates higher robustness and preserves its advantage for all distortion levels.

|  | Bedroom-28 | | | Cat-15 | | | Horse-21 | | |
|---|---|---|---|---|---|---|---|---|---|
| Method | 40 | 20 | 10 | 30 | 20 | 10 | 30 | 20 | 10 |
| DDPM | $49.4 \pm 1.9$ | $46.2 \pm 3.6$ | $38.2 \pm 2.9$ | $53.7 \pm 3.3$ | $49.2 \pm 4.2$ | $42.0 \pm 4.8$ | $65.0 \pm 0.8$ | $63.8 \pm 0.7$ | $56.9 \pm 2.4$ |

Table 4: Evaluation of the proposed method with a different number of labeled training data. Even using less annotated data, DDPM still outperforms most baselines in Table 2.

| Share Train/Test | Share for $t$ | Bedroom-28 | FFHQ-34 |
|---|---|---|---|
| + | + | $49.3 \pm 1.9$ | $59.1 \pm 1.4$ |
| + | - | $49.1 \pm 2.2$ | $59.3 \pm 1.5$ |
| - | - | $48.9 \pm 1.6$ | $59.3 \pm 1.4$ |

Table 5: Performance of the DDPM-based method for different feature extraction variations. All considered stochastic options provide a similar mIoU to the determinstic one.

The results are provided in Table 5. As one can see, the difference in the performance is marginal. We attribute this behavior to the following reasons:

- Our method uses later $t$ of the reverse diffusion process where the noise magnitude is low.
- Since we exploit the deep layers of the UNet model, the noise might not affect the activations from these layers significantly.

**Robustness to input corruptions.** In this experiment, we investigate the robustness of DDPM-based representations. First, we learn pixel classifiers on the clean images using the DDPM, SwAV and MAE representations on the Bedroom-28 and Horse-21 datasets. Then, 18 diverse corruption types, adopted from (Hendrycks & Dietterich, 2019), are applied to test images. Each corruption has five levels of severity. In Figure 6, we provide mean IoUs computed over all corruption types for 1, 3, 5 levels of severity, denoted as "weak", "medium" and "strong", respectively.

One can observe that the proposed DDPM-based method demonstrates higher robustness and preserves its advantage over the SwAV and MAE models even for severe image distortions.

## 5 CONCLUSION

This paper demonstrates that DDPMs can serve as representation learners for discriminative computer vision problems. Compared to GANs, diffusion models allow for a straightforward computation of these representations for real images, and one does not need to learn an additional encoder, which maps images to the latent space. This DDPM's advantage and superior generative quality provide state-of-the-art performance in the few-shot semantic segmentation task. The notable restraint of the DDPM-based segmentation is a requirement of high-quality diffusion models trained on the dataset at hand, which can be challenging for complex domains, like ImageNet or MSCOCO. However, given the rapid research progress on DDPM, we expect they will reach these milestones in the nearest future, thereby extending the range of applicability for the corresponding representations.

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

APPENDIX

## A  EVOLUTION OF PREDICTIVE PERFORMANCE

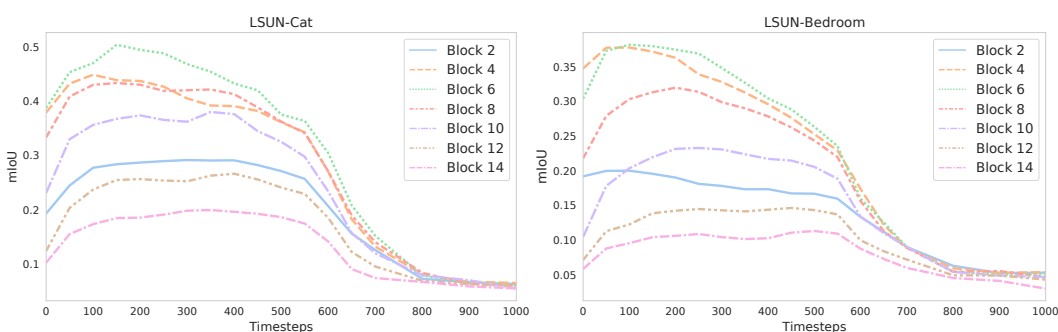

Figure 7: The evolution of predictive performance of DDPM-based pixel-wise representations for different UNet blocks and diffusion steps on LSUN-Cat and LSUN-Bedroom. The blocks are numbered from the deep to shallow ones.

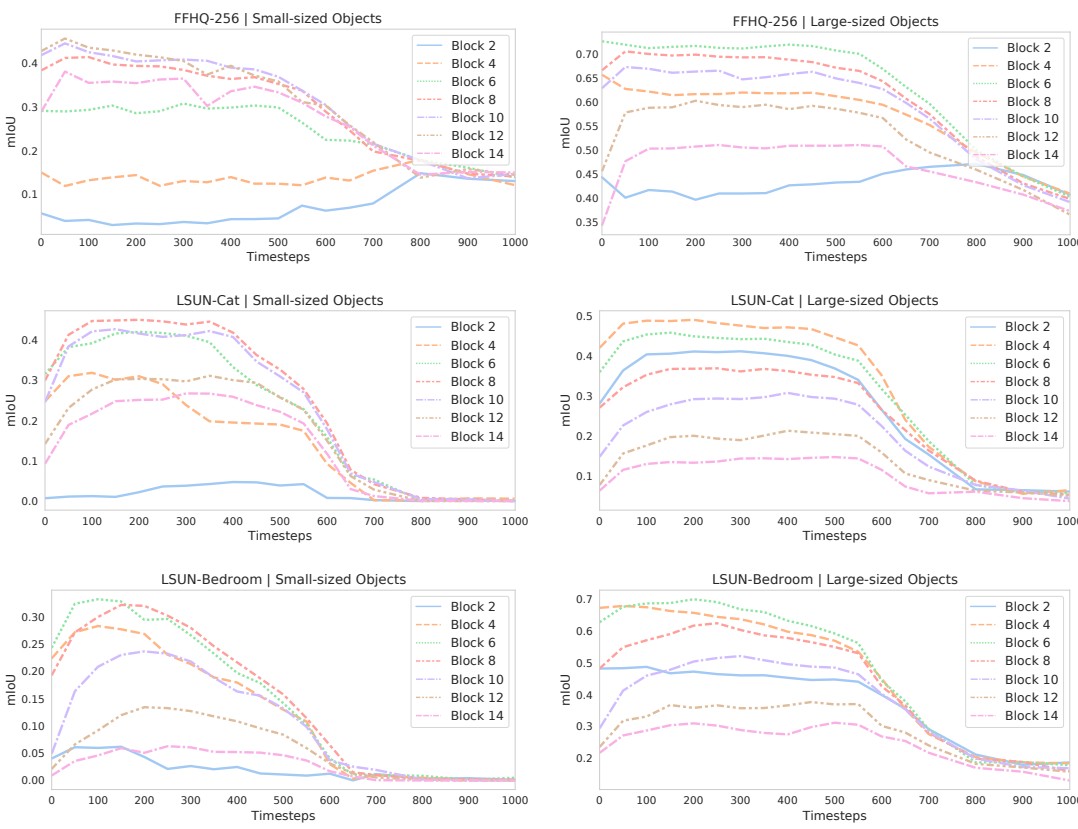

Figure 8: The evolution of predictive performance of DDPM-based pixel-wise representations on the FFHQ-256, LSUN-Cat and LSUN-Bedroom datasets for classes with the smallest (**Left**) and largest (**Right**) average areas.

# B  DATASETDDPM & DATASETGAN SATURATION

| | DatasetDDPM | | | | | DatasetGAN | | | | |
|---|---|---|---|---|---|---|---|---|---|---|
| Dataset | 10k | 20K | 30K | 40K | 50K | 10K | 20K | 30K | 40K | 50K |
| Bedroom-28 | $45.1 \pm 2.3$ | $46.2 \pm 2.3$ | $46.1 \pm 2.8$ | $47.8 \pm 2.3$ | $47.9 \pm 2.9$ | $30.6 \pm 2.3$ | $30.4 \pm 3.1$ | $30.9 \pm 2.4$ | $30.9 \pm 2.4$ | $31.3 \pm 2.7$ |
| FFHQ-34 | $55.9 \pm 0.8$ | $55.8 \pm 0.7$ | $55.9 \pm 0.7$ | $56.0 \pm 0.8$ | $55.9 \pm 0.7$ | $56.4 \pm 1.0$ | $56.9 \pm 1.0$ | $57.0 \pm 1.1$ | $57.0 \pm 1.2$ | $57.0 \pm 1.2$ |
| Cat-15 | $43.6 \pm 3.0$ | $46.4 \pm 1.7$ | $46.2 \pm 1.9$ | $47.4 \pm 1.7$ | $47.6 \pm 1.5$ | $34.7 \pm 2.8$ | $34.8 \pm 2.9$ | $36.3 \pm 2.3$ | $35.8 \pm 2.5$ | $36.5 \pm 2.3$ |
| Horse-21 | $57.0 \pm 1.2$ | $59.5 \pm 0.5$ | $59.0 \pm 2.0$ | $60.4 \pm 1.1$ | $60.8 \pm 0.9$ | $41.6 \pm 2.0$ | $43.1 \pm 1.8$ | $45.4 \pm 1.4$ | $44.5 \pm 1.2$ | $44.6 \pm 1.4$ |

Table 6: Performance of DatasetDDPM and DatasetGAN for $10K{-}50K$ synthetic images in the training dataset. Mean IoU of both methods saturates at $30K{-}50K$ of synthetic data.

# C  TRAINING SETUP

The ensemble of MLPs consists of 10 independent models. Each MLP is trained for $\sim 4$ epochs using the Adam optimizer (Kingma & Ba, 2015) with $0.001$ learning rate. The batch size is $64$. This setting is used for all methods and datasets.

**MLP architecture.** We adopt the MLP architecture from (Zhang et al., 2021). Specifically, we use MLPs with two hidden layers with ReLU nonlinearity and batch normalization. The sizes of hidden layers are $128$ and $32$ for datasets with a number of classes less than $30$, and $256$ and $128$ for others.

Also, we evaluate the performance of the proposed method for twice wider / deeper MLPs on the Bedroom-28 and FFHQ-34 datasets and do not observe any noticeable difference, see Table 7.

| Method | Bedroom-28 | FFHQ-34 |
|---|---|---|
| Original MLP | 49.4 | 59.1 |
| Wider MLP | 49.5 | 59.1 |
| Deeper MLP | 49.3 | 58.9 |

Table 7: Performance of the proposed method for twice wider / deeper MLP architecture within the ensemble. More expressive MLPs do not improve the performance.

# D  PER CLASS IOUS

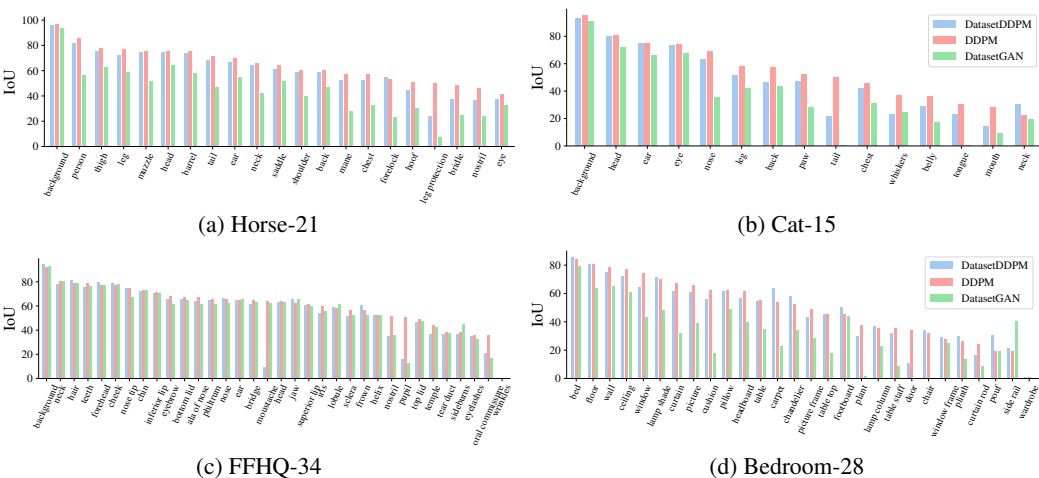

Figure 9: Per class IoUs for DatasetGAN, DatasetDDPM and DDPM.

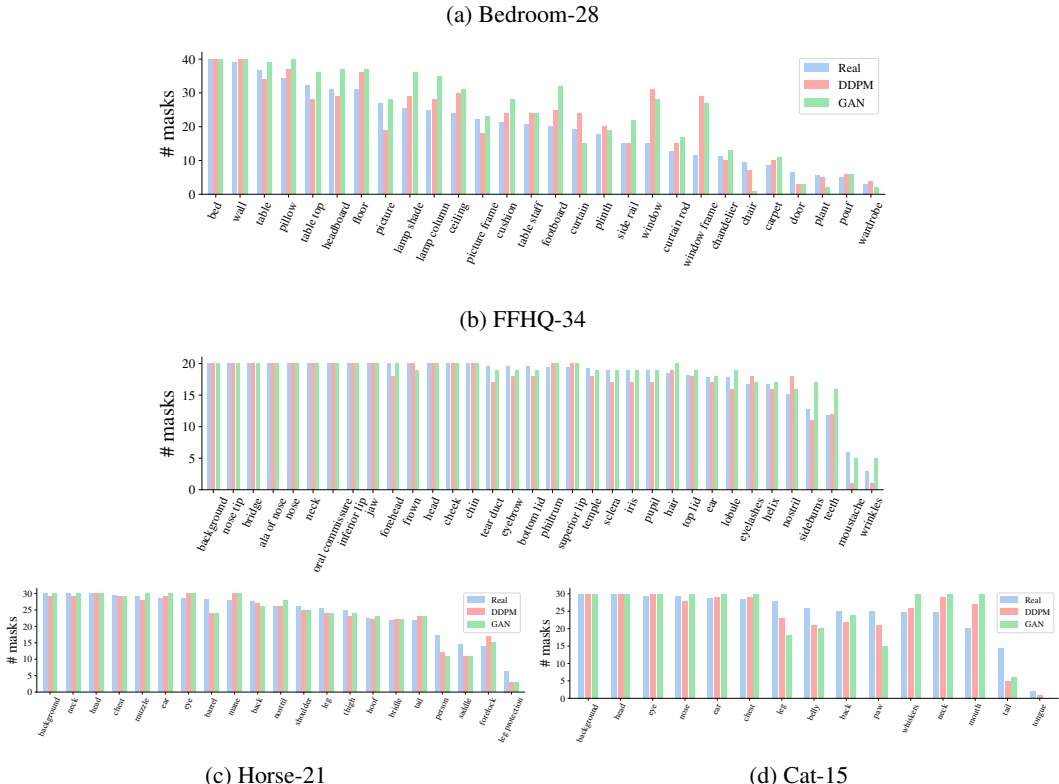

Figure 10: Number of instances of each semantic class in the annotated real and synthetic train sets.

# E    DATASET DETAILS

## E.1    CLASS NAMES

**Bedroom-28**: [bed, footboard, headboard, side rail, carpet, ceiling, chandelier, curtain, cushion, floor, table, table top, picture, pillow, lamp column, lamp shade, wall, window, curtain rod, window frame, chair, picture frame, plinth, door, pouf, wardrobe, plant, table staff]

**FFHQ-34**: [background, head, cheek, chin, ear, helix, lobule, bottom lid, eyelashes, iris, pupil, sclera, tear duct, top lid, eyebrow, forehead, frown, hair, sideburns, jaw, moustache, inferior lip, oral commissure, superior lip, teeth, neck, nose, ala of nose, bridge, nose tip, nostril, philtrum, temple, wrinkles]

**Cat-15**: [background, back, belly, chest, leg, paw, head, ear, eye, mouth, tongue, tail, nose, whiskers, neck]

**Horse-21**: [background, person, back, barrel, bridle, chest, ear, eye, forelock, head, hoof, leg, mane, muzzle, neck, nostril, tail, thigh, saddle, shoulder, leg protection]

**CelebA-19**: [background, cloth, ear_r, eye_g, hair, hat, l_brow, l_ear, l_eye, l_lip, mouth, neck, neck_l, nose, r_brow, r_ear, r_eye, skin, u_lip]

**ADE-Bedroom-30**: [wall, bed, floor, table, lamp, ceiling, painting, windowpane, pillow, curtain, cushion, door, chair, cabinet, chest, mirror, rug, armchair, book, sconce, plant, wardrobe, clock, light, flower, vase, fan, box, shelf, television]

## E.2    CLASS STATISTICS

In Figure 10, we report the statistics of classes computed over annotated real images as well as annotated synthetic images produced by GAN and DDPM.

# F    EXTRACTING REPRESENTATIONS FROM MAE

To obtain pixelwise representations, we apply the model to a fully observed image ($mask\_ratio$=0) of resolution 256 and extract feature maps from the deepest 12 ViT-L blocks . The feature maps from each block have $1024 \times 32 \times 32$ dimensions. Similarly to other methods, we upsample the extracted feature maps to $256 \times 256$ and concatenate them. The overall dimension of the pixel representation is 12288.

In addition, we investigated other feature extraction strategies and got the following observations:

1. Including activations from the decoder did not provide any noticeable gains;
2. Extracting activations right after self-attention layers caused slightly inferior performance;
3. Extracting activations from every second encoder block also provided a bit worse results.

