# OpenReview forum: "Label-Efficient Semantic Segmentation with Diffusion Models"
_ICLR.cc/2022/Conference — ICLR 2022 Poster_

### Official Review · Reviewer_Ck6k · 2021-10-30

**Correctness:** 3
**Technical Novelty And Significance:** 3
**Empirical Novelty And Significance:** 3
**Recommendation:** 8
**Confidence:** 3

**Main Review:**

**Strengths**


* Well motivated

   The problem is well-motivated. The paper _i)_ begins with hypothesizing that the latent activations in the DDPM may contain meaningful semantic information, _ii)_ provides analyses on the behavior of the model (Fig. 1 and Fig. 2), and then _iii)_ provides experiments on multiple datasets, comparing with other generative methods.


* Written clearly

  The paper is easy to follow. It includes sufficient details for understanding the proposed method.


**Weaknesses**


* How would the normalizing flow methods behave in the same setup?

  As one of the generative models, I wonder how the flow-based behaves in this problem setup.
Can a flow-based method also contain meaningful semantic information as DDPM? It would be great if the flow-based model can be also compared in the experiment (if possible) so that it gives more comprehensive comparisons on other types of generative models.


* Experiment setup / datasets

  The number of test images with annotations is certainly small (ref. Table 1). I am not so sure if the conclusions that are drawn from the small-scale test set can be consistent and generalizable to the large scale. For the experiment, I wonder if it's possible to use any kinds of popular, large-scale semantic segmentation datasets that already contain the annotations, for example, NYUv2, Cityscapes, or ScanNet.
I am not saying that the paper has to provide additional experiments on "those" datasets. Rather, the experiment on the small-scale setups is not so ideal, so I wonder if it's possible to conduct the experiment on such kinds of datasets with a sufficient number of annotated images.



**Summary Of The Paper:**

The paper demonstrates that the intermediate activations in Denoising diffusion probabilistic models (DDPM) can capture semantic information and thus can serve as representation for high-level vision tasks.
The paper provides an interesting analysis of how well each layer at each diffusion step in DDPM can serve as a representation for semantic segmentation (Fig. 1 and Fig. 2).
Also, through small-scale yet valid experiments, the method shows the learned intermediate activations from DDPM contain semantic cues well, and better than other generative approaches.

**Summary Of The Review:**

The paper certainly draws a very interesting question, "Can DDPM serve as a representation learner for high-level computer vision tasks?", and provides insightful analyses and experiments.
However, I have a concern if the conclusions drawn from the provided small-scale experiment can be valid for a large-scale/general setup.
For now, I want to be a bit conservative in the rating and possibly increase based on the discussion and other reviews.

----

**After the author response**:
The rebuttal resolved my main concerns (small-scale experiments). The other reviewers also claimed valid concerns, and they seem resolved/answered well (providing ablation studies, more details, and justifications). Thus, I raise my rating to Accept.

---

> ### Author Response · Authors · 2021-11-16
> **Response to Reviewer Ck6k**
>
> Thank you for your review. We address your concerns below.
>
> ---
>
> **Q1: How would the normalizing flow methods behave in the same setup?**
>
> ---
>
> To the best of our knowledge, the only NF-based models applicable to the reasonable resolutions are Glow-based: Glow[1], WaveletFlow[2]. These architectures produce samples in a different way than convolutional ones. Since NFs have to be invertible, they use an unsqueeze operation for upsampling. Unlike the pooling operation, this one splits the channel dimension of the intermediate activations (CxHxW) by 4 and stacks the patches to the spatial dimension to obtain the size (C/4 x 2*H x 2*W). This means that opposed to convolutional models, the (i, j) pixel of the feature map does not correspond to the (i, j) patch of the input image. Therefore, it is not clear how to obtain the pixel-wise representations in this case.
>
> In addition, NFs have significantly lower sample fidelity compared to GANs or DDPMs on the considered datasets. Therefore, we believe that it would be more relevant to consider, e.g., autoencoder-based models. We added recent VDVAE[3] and ALAE[4] to the table in the response above. On FFHQ-256, VDVAE is comparable to DDPM, however, there is no evidence that is able to provide reasonable performance on LSUN.  To the best of our knowledge, ALAE is the only autoencoder-based model applied to LSUN with 256 resolution and has a publicly available checkpoint on LSUN-Bedroom.  We evaluate ALAE on ADE-Bedroom  and LSUN-Bedroom and observe that it is significantly inferior to DDPMs.
>
> Please, let us know if you would like us to evaluate any other publicly available model pretrained on the considered datasets. We would be happy to add it to the comparison.
>
> ---
>
> **Q2: Is it possible to use any kinds of popular, large-scale semantic segmentation datasets?**
>
> Yes, we agree that the dataset with a reasonable number of annotated images can make our assumptions stronger. We collect the ADE-Bedroom dataset from ADE20k with the evaluation set of 650 images and provide the results in our response above. DDPM still significantly outperforms the baselines.
>
> ---
>
> **?** Have we addressed your concerns? Are there any other questions that could affect your score?
>
> ---
>
> [1] Kingma, Diederik P., and Prafulla Dhariwal. "Glow: Generative flow with invertible 1x1 convolutions." arXiv preprint arXiv:1807.03039 (2018).
>
> [2] Yu, Jason J., Konstantinos G. Derpanis, and Marcus A. Brubaker. "Wavelet flow: Fast training of high resolution normalizing flows." Advances in Neural Information Processing Systems 33 (2020): 6184-6196.
>
> [3] Child, Rewon. "Very deep vaes generalize autoregressive models and can outperform them on images." arXiv preprint arXiv:2011.10650 (2020).
>
> [4] Pidhorskyi, Stanislav, Donald A. Adjeroh, and Gianfranco Doretto. "Adversarial latent autoencoders." Proceedings of the IEEE/CVF Conference on Computer Vision and Pattern Recognition. 2020.

---

> > ### Comment · Reviewer_Ck6k · 2021-11-18
> > **Response to the rebuttal**
> >
> > Thank you for the response and the additional experiments on datasets with more test images.
> > The experiment setup (e.g., the number of test images) looks fine: I guess no additional experiment is needed.
> > I will closely look into the response as well as other reviews and summarize my points by tomorrow.
> > Thank you!

---

> > ### Comment · Reviewer_Ck6k · 2021-11-19
> > **Response to the rebuttal**
> >
> > I carefully read the responses as well as the other reviews.
> > The additional evidence and clarification resolve my main concerns, as well as those from the other reviews (such as hyper-parameters, experiment protocol, additional ablation study, etc.).
> >
> > -  The usage of more test images (20 -> 650) makes the conclusion of the paper more solid, which is good!
> >
> > -  The revised draft looks also good: it includes the additional experiments and some corrections from the other reviewers. By the way, (probably in the supplementary material), it would be great if the paper also includes the additional ablation studies that are demonstrated here on the review page (i..e, width-depth of MLP, # of synthetic data, ...). I think it will help readers understand the paper much better.
> >
> > - One last question: the accuracy of the DatasetGAN is quite improved, compared to the previous version. Is it due to the bug fix or the usage of more training images?
> >
> > In summary, the response resolves my main concerns. I will update my rating and reviews while waiting for any comments from the other reviewers.

---

> > > ### Author Response · Authors · 2021-11-19
> > > **Response to Reviewer Ck6k (#2)**
> > >
> > > Thanks for your response! We are happy that we could address your concerns.
> > >
> > > * (Include some experiments in the revision) We will place those experiments in the appendix. We are going to update it in a few days.
> > >
> > > * (DatasetGAN) The bug fix caused the significant improvement on the FFHQ (\~12%) and LSUN-cat (\~9%) datasets. Increasing the amounts of training data led to additional ~2-4% for all datasets.

---

> > > ### Author Response · Authors · 2021-11-23
> > > **Revised paper**
> > >
> > > We've uploaded the final revision. We've included the ablation studies from the rebuttal and added more details and clarifications throughout the paper for better understanding. Thanks for your comments and suggestions!

---

### Official Review · Reviewer_s8q7 · 2021-11-02

**Correctness:** 3
**Technical Novelty And Significance:** 3
**Empirical Novelty And Significance:** 3
**Recommendation:** 8
**Confidence:** 4

**Main Review:**

Strengths:
- The paper is clearly written and has no significant missing details. Although the code is not provided, I do not see this as a major weakness, as the DDPM training and some models are borrowed from [Dhariwal and Nichol, 2021].
- The experiments and baselines are well-chosen. Figure 2 (unsupervised clustering of deep features) is excellent motivation for the proposed method, and the analysis (S.4.1, Figures 4 and 5) answered many of my initial questions about the results.
- Most importantly, this work can lead to further research on DDPMs as feature extractors. The idea is potentially applicable to any computer vision problems where other kinds of pretrained features (e.g., VGG) are used. As the authors note in the conclusion, the line of work begun by this paper can find increasing uses as DDPMs become more powerful.

Weaknesses / questions:
- Significance of the results in Table 2: The data is small -- as few as 20 training and evaluation images -- and does not use a validation set for parameter search, but there are parameter choices that are not motivated (layers and time steps, number of models in the ensemble of MLPs). The claim that DDPM outperforms baselines would be more convincing if results were averaged over many runs, or at least a k-fold cross-validation were done.
- If my understanding of Figure 3 is correct, the feature extraction is stochastic, as it depends on the noise applied to the image to produce the input to the U-Net. An analysis of how the randomness in features affects training and evaluation would be beneficial. (Were the same features for each image used for training each MLP?)
- The hierarchy of results DDPM > DatasetDDPM > DatasetGAN is less surprising than DDPM > Self-Supervised Pretrain. For the latter, only one self-supervised model and choice of layers was used. Were other unsupervised feature extractors considered? For example, do features from an autoencoder and from a DDPM of the same architecture capture similar high-level information?

**Summary Of The Paper:**

This paper proposes to treat deep activations of denoising diffusion probabilistic models trained on image datasets as unsupervised pixel features. To extract features for an image, one performs a fixed number of steps of the diffusion process, passes the resulting noisy image through the U-Net denoiser, and upsamples activations from a chosen layer. These pixel representations are used as inputs to simple classifiers that perform semantic segmentation. This approach is validated on three image datasets, where it shows strong results compared to baselines in a few-shot setting.

**Summary Of The Review:**

The paper presents a new use of DDPMs as unsupervised representation learners. Although I see the small-scale experiments as more exploratory than competitive, it contributes to our understanding of an increasingly popular modeling approach.

---

> ### Author Response · Authors · 2021-11-16
> **Response to Reviewer s8q7**
>
> Thank you for your detailed comments! We thoroughly address them below.
>
> ---
>
> **Q1: An analysis of how the randomness in features affects training and evaluation would be beneficial.**
>
> We agree that it is an interesting point for analysis. In our experiments, we share the same noise sample during the training and evaluation for all timesteps t.  Below, we also consider two alternative options:
>
> 1) Different noise samples for different t but share them for training and evaluation.
> 2) Different noise everywhere
>
> The results are presented in the table below. As one can see, the difference in performance is marginal.
>
> **LSUN-bedroom**
>
> | Shared for train/test | Shared for different t | mIoU    	|
> |:-------------------- |:--------------------- |:----------- |
> | +                	| +                 	| 46.1 +- 1.9 |
> | +                	| -                 	| 45.8 +- 1.4 |
> | -                	| -                 	| 45.6 +- 1.8 |
>
> **FFHQ-256**
>
> | Shared for train/test | Shared for different t | mIoU    	|
> |:----------------------- |:---------------------- |:----------- |
> | +                   	| +                  	| 57.0 +- 1.4 |
> | +                   	| -                  	| 57.2 +- 1.6 |
> | -                   	| -                  	| 57.1 +- 1.4 |
>
> ---
>
> **Q2: There are parameter choices that are not motivated.**
>
> In all our experiments, we take the same training and evaluation protocols, number of models, and other hyperparameters as in datasetGAN [1]. In order to avoid overfitting on our datasets,  we select the DDPM layers and time steps roughly according to the general behaviour on Figure 1 and freeze them for all other datasets.
>
> ---
>
> **Q3: The claim that DDPM outperforms baselines would be more convincing if results were averaged over many runs, or at least a k-fold cross-validation were done.**
>
> We agree on this. We have already addressed this in the updated table above.
>
> ---
>
> **Q4: The hierarchy of results DDPM > DatasetDDPM > DatasetGAN is less surprising than DDPM > Self-Supervised Pretrain. For the latter, only one self-supervised model and choice of layers was used. Were other unsupervised feature extractors considered? For example, do features from an autoencoder and from a DDPM of the same architecture capture similar high-level information?**
>
> We agree that adding other unsupervised features extractors would be beneficial. We provide the comparison to VDVAE [2] and ALAE [3] in the recently released table in our response above.  VDVAE demonstrates comparable performance on FFHQ-256, while ALAE is inferior to most other methods on both LSUN-Bedroom and FFHQ-256. Note, that it is not generally clear if autoencoders can reach GAN/DDPM generative performance on the complex LSUN datasets.
>
> ---
>
> [1] Zhang, Yuxuan, et al. "Datasetgan: Efficient labeled data factory with minimal human effort." Proceedings of the IEEE/CVF Conference on Computer Vision and Pattern Recognition. 2021.
>
> [2] Child, Rewon. "Very deep vaes generalize autoregressive models and can outperform them on images." arXiv preprint arXiv:2011.10650 (2020).
>
> [3] Pidhorskyi, Stanislav, Donald A. Adjeroh, and Gianfranco Doretto. "Adversarial latent autoencoders." Proceedings of the IEEE/CVF Conference on Computer Vision and Pattern Recognition. 2020.

---

> ### Author Response · Authors · 2021-11-19
> **Response to Reviewer s8q7 (#2)**
>
> We would like to inform you that we have recently uploaded the revision of the main paper.
>
> Could you please let us know whether we need to conduct any additional experiments to address your questions? There is not much time left until the end of the discussion period, so we would like to be sure that we can provide all necessary numbers by November 22.
>
> Looking forward to hearing from you!

---

> > ### Comment · Reviewer_s8q7 · 2021-11-20
> > **Response to the authors**
> >
> > Thank you for your detailed response. The new results (a substantial effort!) and clarifications strengthen the paper's main claims and address many of my concerns. I would like to see the paper accepted and have updated the score accordingly.

---

### Official Review · Reviewer_LFKA · 2021-11-02

**Correctness:** 4
**Technical Novelty And Significance:** 3
**Empirical Novelty And Significance:** 3
**Recommendation:** 6
**Confidence:** 4

**Main Review:**

I enjoyed reading this paper. While relatively straightforward and obvious enough, the question of whether DDPMs are good representation learners is definitely interesting. The positive result in this paper will be of broad interest among the growing community working on generative models. One may wonder how well the representations transfer to other dense vision tasks, like instance segmentation, or monocular depth for example. The paper would have been much stronger had the authors shown similarly strong transfer on such related or complementary tasks.

The paper is well written. Nevertheless, I have several small suggestions that might improve the clarity of the paper:
- A diagram or table for the UNET showing layer numbering, layer down-sampling rates, number of channels per layer, etc may be useful.  In particular it may not be obvious to readers how to associate the number of a given layer with network depth and sampling rate, as these affect the scale of the image features.
- Section 3.1 talks about 'later' or 'earlier' diffusion steps (associated with small and large values of t, respectively). But small values of t are more naturally associated with early stages of the diffusion process, and the later stages of the generation (or reverse diffusion) process.  So when talking about the later steps of the diffusion process in Section 3.1 (small t), I would instead talk about the later stages of the reverse diffusion process.
- Although not explicitly stated, I assume the MLP is applied to the feature vector at each pixel location independent of neighboring pixel locations? It might also help readers to state the dimension of the feature vector used as input to infer the pixel label.
- In Section 4 on datasets, when talking about baseline methods, the paper states that "Methods use the same number of annotated images for training and the same set of images for evaluation."  I assume that the different methods use the same images for training, not just the same number of images?

The experimental results are clear, but they also raise many questions. For example,
- To what extent do results shown in Figure 1 depend on the size of the training set?
- How does segmentation performance depend on the width and depth of the MLP for pixel label prediction?
- In discussing model results of datasetDDPM vs DatasetGAN, the paper explains that the better performance of DatasetDDPM is due to higher quality image generation by DDPMs. Could the difference due to mode dropping?
- In Table 2, self-supervised pretraining outperforms supervised pretraining. Some discussion of this would be useful.
- With the use of synthetic data, by datasetGAN and DatasetDDPM, how do the results vary as a function of the amount of synthetic data.  While performance is generally higher when trained on real data, can one close this gap with more synthetic data?
- Table 5 shows dependence of segmentation performance on size of the labelled training set. Does performance continue to rise as the amount of labelled data increases

**Summary Of The Paper:**

This paper explores to what extent Denoising Diffusion Probabilistic Models (DDPMs) serve as good representational learners for transfer or semi-supervised learning on downstream tasks. They are particularly interested in the semantic segmentation as a prototypical dense computer vision task.  They find that DDPM do provide useful representations.  In particular they show that the middle U-NET layers, at intermediate steps in the reverse diffusion process produce activations from which a simple MLP can infer good segmentations results (as measured by IoU).  They demonstrate very good results in a substantial number of experiments in which a DDPM is learned from a large set of unlabeled images, and then the MLP is trained on a relatively small set of labelled in-domain images, and then evaluated. They show that a simple segmentation network trained on top of DDPM activations outperforming a wide array of baseline models, including DatasetGAN and other recent SOTA methods.


**Summary Of The Review:**

This is a well written paper with an interesting result on the use of DDPMs for representation learning.  This is particularly interesting because there is widespread interest in diffusion models at present, and the question of DDPMs for representation learning has not been addressed to the best of my knowledge.  That said, the approach taken in the paper is relatively straightforward, and only one downstream task is considered, albeit with extensive baselines.  It would have been great to see results on another task.

I also want to thanks for the reviewers for their extensive responses to questions raised by the reviewers. Most of my questions have been addressed, and the new results reported support the results reported in the original submission.  I continue to believe that this paper is above the bar and would make a good addition to the conference.  My rating of 6 should be interpreted as 6+ or 7.

---

> ### Author Response · Authors · 2021-11-16
> **Response to Reviewer LFKA**
>
> Thank you for your review. We provide our comments on the questions below.
>
> ---
>
> **Q1: To what extent do results shown in Figure 1 depend on the size of the training set?**
>
> We conducted the same experiment for the twice smaller training sets and observed the similar behavior. Please, let us know if you need the empirical confirmation (translating those plots into text is quite time-consuming).
>
> ---
>
> **Q2: How does segmentation performance depend on the width and depth of the MLP for pixel label prediction?**
>
> We conducted experiments for twice wider / deeper MLPs on the LSUN-Bedroom and FFHQ-256 datasets and did not observe any noticeable difference:
>
> | LSUN-bedroom | mIoU |
> |:------------ |:---- |
> | Original MLP   | 46.1 |
> | Wider MLP	| 46.0 |
> | Deeper MLP   | 46.1 |
>
> | FFHQ-256   | mIoU |
> |:---------- | ---- |
> | Original MLP | 57.0 |
> | Wider MLP  | 57.0 |
> | Deeper MLP | 56.9 |
>
> ---
>
> **Q3: The paper explains that the better performance of DatasetDDPM is due to higher quality image generation by DDPMs. Could the difference be due to mode dropping?**
>
> In the paper, we meant the general higher quality of DDPMs compared to GANs both in terms of realism and coverage (see Precision and Recall values in Table 5 [1]). Our current understanding is that both higher Precision and Recall contribute to the advantage of DatasetsDDPM over DatasetGAN.
>
> ---
>
> **Q4: In Table 2, self-supervised pretraining outperforms supervised pretraining. Some discussion of this would be useful.**
>
> We consider the “supervised pretraining” baseline since it was used in the DatasetGAN paper [2]. This baseline is pretrained on the MSCOCO dataset, which classes differ from the classes in our benchmarks. We attribute the poor performance of the “supervised pretraining” baseline to this domain gap.
>
> ---
>
> **Q5: How do the results of  datasetGAN and DatasetDDPM vary as a function of the amount of synthetic data. While performance is generally higher when trained on real data, can one close this gap with more synthetic data?**
>
> In our experiments, we use 10k synthetic images for datasetDDPM/datasetGAN and remove 10% of samples with most uncertain predictions according to [2]. Below, we provide the results for synthetic datasets of 20k, 30k, 40k and 50k (-10% of removed samples).
>
> |             	| LSUN-bedroom | LSUN-Cat | LSUN-Horse |
> |:--------------- |:------------------- |:--------------- |:----------------- |
> | DDPM        	| 46.1 +- 1.9     	| 52.3 +- 3.0 	| 63.1 +- 0.9   	|
> | datasetDDPM 10k | 43.8 +- 3.5     | 42.9 +- 2.8 	| 55.6 +-0.9    	|
> | datasetDDPM 20k | 45,6 +- 3.6     | 43.2 +- 2.9 	| 58.1 +- 0.8   	|
> | datasetDDPM 30k | 45.3 +- 2.7     | 43.5 +- 2.4 	| 59.5 +- 1.0   	|
> | datasetDDPM 40k | 46.9 +- 2.5     | 44.3 +- 2.6     | 60.4 +- 1.2    |
> | datasetDDPM 50k | 46.8 +- 3.0     | 45.4 +- 2.8     | 59.6 +- 1.1    |
> | datasetGAN 10k | 30.6 +- 2.3     | 34.7 +- 2.8 	| 41.6 +- 2.0     |
> | datasetGAN 20k | 30,4 +- 3.1     | 34.8 +- 2.9 	| 43.1 +- 1.8   	|
> | datasetGAN 30k | 30.9 +- 2.4     | 36.3 +- 2.3 	| 45.4 +- 1.4   	|
> | datasetGAN 40k | 30.8 +- 2.5     | 35.8 +- 2.5     | 44.5 +- 1.2    |
> | datasetGAN 50k | 31.3 +- 2.7     | 36.5 +- 2.3     | 44.6 +- 1.4    |
>
> ---
>
> **Q6: Table 5 shows dependence of segmentation performance on size of the labelled training set. Does performance continue to rise as the amount of labelled data increases?**
>
> While additional annotation is very expensive, we provide the results for the ADE Bedroom dataset. We consider the dataset with 30 classes  and increase the size of the training set from 50 to 70 and 86, respectively.
>
> **ADE Bedroom**
>
> |	Train size	| 	mIoU  	|
> |:-----------------|:---------------------|
> |      	50     	|   	33.7  	|
> |      	70     	|  	35.4   	|
> |      	86     	|  	36.9   	|
>
>
> The table above shows that more training data corresponds to higher performance, as expected.
>
> ---
>
> **Q7: The paper states that "Methods use the same number of annotated images for training and the same set of images for evaluation." I assume that the different methods use the same images for training, not just the same number of images?**
>
> Yes, exactly. We will reformulate this in the revision.
>
> ---
>
> **?** We hope that we have addressed all your concerns and questions. Is there anything else that could affect your evaluation?
>
> ---
> [1] Dhariwal, Prafulla, and Alex Nichol. "Diffusion models beat gans on image synthesis." arXiv preprint arXiv:2105.05233 (2021).
>
> [2] Zhang, Yuxuan, et al. "Datasetgan: Efficient labeled data factory with minimal human effort." Proceedings of the IEEE/CVF Conference on Computer Vision and Pattern Recognition. 2021.

---

> ### Author Response · Authors · 2021-11-19
> **Response to Reviewer LFKA (#2)**
>
> We would like to inform you that we have recently uploaded the revision of the main paper.
>
> Since it is three days until the end of the discussion period, we would be happy to know whether we have addressed your concerns and need to conduct any additional experiments?
>
> Looking forward to hearing from you!

---

### Author Response · Authors · 2021-11-16
**Experimental update**

First of all, we thank the reviewers for their time and valuable comments. Before we reply to them individually and address their concerns, we would like to present an update in our experiments:

1) Added new datasets:

  * ADE Bedroom ---  a part of the ADE20k dataset [1] that consists of 700 annotated bedroom images with 30 most frequent object categories. The train set has 50 images. The evaluation set - 650.

  * LSUN-Horse --- We manually annotate 30/30/30 real/ddpm/gan images for training and 30 real images for evaluation. The number of classes is 21.

2) Evaluated all methods for 5 runs on the different data splits and added corresponding mean and standard deviations.

3) Enhanced SSL baseline by training SwAV on the corresponding datasets. The input resolution during the training is 256, similar to DDPM.

4) Added autoencoder-based baselines. We evaluate publicly available VDVAE[2] and ALAE[3] checkpoints. VDVAE is trained on FFHQ-256. ALAE has checkpoints on the LSUN-Bedroom and FFHQ-1024.

5) Increased the quality of image annotations, which led to consistent improvements on FFHQ-256 and LSUN-Cat for all methods.

6) Added datasetGAN and datasetDDPM trained on the larger amounts of synthetic data. We increase the number of samples until the performance on the validation set is not saturated.

7) Fixed a bug in the datasetGAN configuration for FFHQ-256 and LSUN-Cat datasets. It became comparable to other methods on FFHQ-256 and closer to datasetDDPM on LSUN-Cats.

The overall table is provided below:

| Method       | LSUN-Bedroom | FFHQ-256 	| LSUN-Cat | LSUN-Horse  | ADE Bedroom 30 |
|:------------- |:-------------- |:--------------- |:--------------- |:--------------- |:--------------- |
| ALAE [1]  	| 20.0 +- 1.0     |  48.1 +- 1.3  	| -- 	| --          	| 15.0 +- 0.5      |
| VDVAE [2] 	| --         	| **57.3 +- 1.1**    | -- | --          	| --          	|
| Gan Inversion  | 13.9 +- 0.6 	| 51.7 +- 0.8 	| 21.4 +- 1.7 	| 17.7 +- 0.4 	| 11.1 +- 0.2 |
| SwAV      	 | 41.0 +- 2.3 	| 55.4 +- 0.6 	| 44.1 +- 2.1 	| 51.7 +- 0.5 	| 30.3 +- 1.5 |
| DatasetGAN	 | 31.3 +- 2.7 	| **57.0 +- 1.0** | 36.5 +- 2.3 	| 45.4 +- 1.4 	| --                |
| DatasetDDPM (Ours)  | **46.9 +- 2.8** |  56.0 +- 0.9    | 45.4 +- 2.8 	| 60.4 +- 1.2 	| --              |
| DDPM      	(Ours)  | **46.1 +- 1.9** | **57.0 +- 1.4** | **52.3 +- 3.0** | **63.1 +- 0.9** | **32.3 +- 1.5** |



[1] Zhou, Bolei, et al. "Semantic understanding of scenes through the ade20k dataset." International Journal of Computer Vision 127.3 (2019): 302-321.

[2] Child, Rewon. "Very deep vaes generalize autoregressive models and can outperform them on images." arXiv preprint arXiv:2011.10650 (2020).

[3] Pidhorskyi, Stanislav, Donald A. Adjeroh, and Gianfranco Doretto. "Adversarial latent autoencoders." Proceedings of the IEEE/CVF Conference on Computer Vision and Pattern Recognition. 2020.

---

### Decision · Program_Chairs · 2022-01-20

**Decision:**

Accept (Poster)

**Comment:**

The paper proposes using the intermediate representation learned in a denoising diffusion model for the label-efficient semantic segmentation task. The reviewers are generally positive with the submission. They like the simplicity of the proposed algorithm. They also like the effort of the paper in verifying the intermediate representation learned by a diffusion model is semantically meaningful and can be used for segmentation. Initially, there was some concern about the size of the validation set, which is addressed by the rebuttal. Consolidating the reviews and rebuttals, the meta-reviewer agrees with the assessment of the reviewers and would like to recommend acceptance of the paper.